# *PUMA* gene delivery to synoviocytes reduces inflammation and degeneration of arthritic joints

Saw-See Hong[1,2,3], Hubert Marotte [4], Guillaume Courbon[4], Gary S. Firestein[5], Pierre Boulanger[1,2] & Pierre Miossec[1,6]

In rheumatoid arthritis (RA), the proliferation of fibroblast-like synoviocytes (FLS) is the cause of chronic inflammation in joints and of joint damage. Delivery of the pro-apoptotic gene *PUMA* to FLS via human adenovirus type 5 (HAdV5) vectors has been tested as a therapeutic approach, but efficiency is hampered by low transduction, as FLS do not express HAdV5 receptors on the cell surface. Here we show that efficient transduction of PUMA in FLS can be achieved by conjugating HAdV5 to a baculovirus, which binds to the cell surface via the envelope glycoprotein Gp64. Intra-articular injection in an adjuvant-induced rat model of RA induces apoptosis of FLS, leading to significant decrease in joint inflammation, joint damage, and bone loss with improvement in joint function and mobility. Our results demonstrate the therapeutic potential of *PUMA* gene therapy as a local treatment in various forms of arthritis in which abnormal FLS proliferation is implicated.

[1] University of Lyon, Lyon 69007, France. [2] UMR754-INRA-EPHE, Unit of Viral Infections and Comparative Pathology, University of Lyon, Lyon 69007, France. [3] INSERM, 101 rue de Tolbiac, Paris 75654 Cedex 13, France. [4] Department of Bone and Osteoarticular Biology INSERM U1059, University Jean Monnet University of Lyon, Saint-Étienne 42100, France. [5] School of Medicine, Division of Rheumatology, Allergy and Immunology, University of California at San Diego, La Jolla, CA 92093, USA. [6] Unit of Immunogenomics and Inflammation EA 4130, Department of Clinical Immunology and Rheumatology, Hôpital Edouard Herriot, University of Lyon, Lyon 69437, France. Correspondence and requests for materials should be addressed to S.-S.H. (email: saw-see.hong@univ-lyon1.fr) or to P.M. (email: pierre.miossec@univ-lyon1.fr)

Rheumatoid arthritis (RA) is the prototype of polyarticular inflammatory disease, affecting ~1% of the world population. Other forms of arthritis specifically in children affect a single or very few joints. Pigmented villonodular synovitis (PVNS) is a tumour that occurs inside the synovial membrane, with a high tendency of recurrence despite surgery. The use of biological drugs has been a major advance for the treatment of RA. However, ~30% of RA patients do not respond to these drugs, which are expensive and can cause severe side-effects[1, 2]. Intra-articular treatment with radio-isotopes for instance, has been effective in RA and PVNS but has major restrictions related to the use of radio-active material. There is therefore a necessity for improvement or alternatives in the local treatment of arthritis.

In the inflamed joint, the uncontrolled proliferation and accumulation of fibroblast-like synoviocytes (FLS) are the main cause of chronic inflammation and its progression to joint damage[3, 4]. This results in part from acquired molecular changes in FLS leading to reduced sensitivity to cell death signals. Apoptosis-inducing strategies targeting FLS have been considered as treatment of arthritis[5–7]. *In vitro* experiments using a plasmid vector to express the proapoptotic gene *PUMA* (p53 upregulated modulator of apoptosis) in FLS, showed the efficacy of PUMA in inducing cell apoptosis[8, 9], a phenomenon which was independent of the p53 status of the synovium[9]. These preliminary data suggested that the strategy of PUMA-induced apoptosis in FLS could block the hyperplasia of the synovial intimal lining.

A variety of non-viral and viral vectors have been tested for the local and systemic treatment of rheumatic diseases by gene therapy[10]. The human adenovirus type 5 (HAdV5)-based vectors gave the best results, despite low efficiency in transduction of rheumatoid synovium in RA animal models[11]. HAdV5 infection is initiated by the attachment of the viral vector to its high-affinity receptor, the Coxsackie-adenovirus receptor (CAR), on the surface of cells[12]. However, human FLS do not express CAR on their surface and are thus poorly transduced by HAdV5 vectors[13].

To overcome this problem of vector inefficiency, we design a novel gene delivery strategy, in which HAdV5-PUMA was 'piggybacked' on a baculovirus vector carrying CAR on its envelope[14], resulting in the efficient cell entry of the vector BV^CAR HAdV5-PUMA into the FLS. We demonstrate in this study that *PUMA* gene transfer into FLS by BV^CAR HAdV5-PUMA results in rapid and extensive cell death by PUMA-induced apoptosis. The pro-apoptotic effect is not substantially reduced in the presence of proinflammatory cytokines, which mimic the environment of inflamed joints. Using the adjuvant-induced arthritis (AIA) rat model, we find that a single intra-articular injection of BV^CAR HAdV5-PUMA significantly decreases joint inflammation, and improves joint function with reduced joint damage and bone loss. The results of this study show that the intra-articular administration of a PUMA-expressing vector has therapeutic potential as a treatment for various forms of arthritis in which FLS proliferation is implicated.

## Results

**Efficient transduction of FLS by BV^CAR HAdV5 vector complex.** The use of HAdV5 as a gene transfer vector for FLS has been limited due to their non-permissiveness to HAdV5 as they do not express CAR, the cellular receptor of HAdV5 on their surface. To overcome this hurdle, we design an efficient gene delivery strategy using a dual vector system by complexing HAdV5 to a baculovirus vector carrying CAR molecules on its envelope (BV^CAR; Fig. 1a,b)[14]. The BV^CAR HAdV5 complex binds to the cell surface via the baculoviral envelope glycoprotein Gp64 (Fig. 1a,b), and is internalized by endocytosis, even in cells lacking

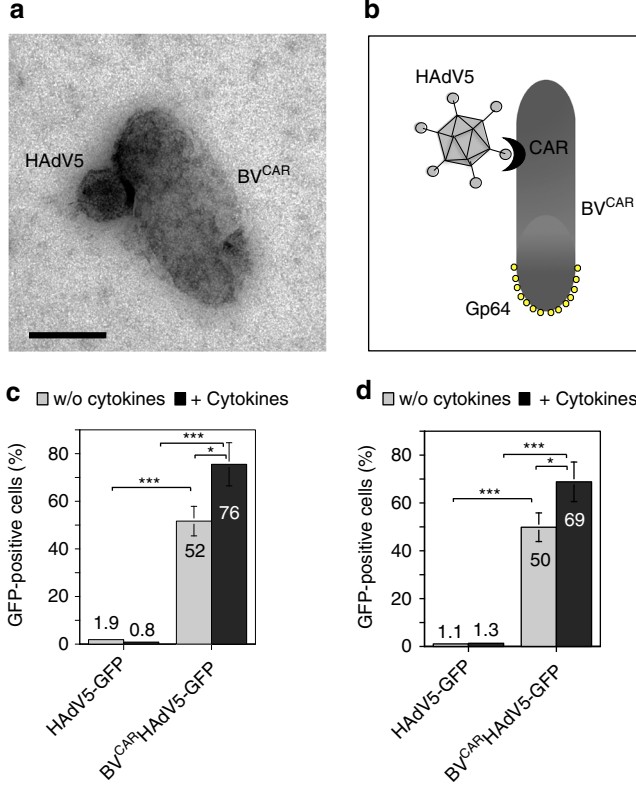

**Fig. 1** Vector duo and transduction of human and rat synoviocytes. **a** The BV^CAR HAdV5 binary complex can be visualize by electron microscopy. The HAdV5 vector with a icosahedral shape, is bound to BV^CAR, a rod-shaped baculovirus which has incorporated the CAR glycoproteins in its envelope. Scale bar, 100 nm. **b** The schematic representation shows the BV^CAR HAdV5 binary complex, with HAdV5 bound to BV^CAR via the CAR glycoprotein at the baculoviral membrane. The baculoviral envelope glycoproteins Gp64 are localized at the head of the baculoviral particle. **c** Human and **d** rat fibroblast-like synoviocytes (FLS) non-treated (*grey bars*) or pre-treated (*black bars*) with the proinflammatory cytokines were transduced with HAdV5-GFP alone, or with the BV^CAR HAdV5-GFP complex. The figures represent the percentage of GFP-positive FLS cells analysed by flow cytometry. The average of three independent experiments, each in triplicates are shown. Statistical analysis using one-way ANOVA test, ***$P < 0.001$, **$P < 0.01$, *$P < 0.05$

CAR, such as FLS[14]. BV^CAR serves as a transporter to bring the HAd5V vector into the target cells. Transduction of human and rat FLS with HAdV5-GFP alone, at a vector dose of 20 vector particles per cell (vp/cell), gave <2% of GFP-positive cells (Fig. 1c, d; grey bars), whereas more than 50% GFP-positive cells were obtained with the BV^CAR HAdV5-GFP complex at the same vector dose (Fig. 1c,d; grey bars). To mimic the inflammatory conditions in arthritic joints, FLSs were pre-treated with proinflammatory cytokines (TNFα, IL-17) before transduction with BV^CAR HAdV5-GFP. This resulted in almost 70% GFP-positive cells (Fig. 1c,d; black bars).

**Cell death of FLS transduced by the BV^CAR-HAdV5-PUMA vector.** An earlier study shows that transfection of FLSs with a plasmid expressing PUMA resulted in rapid apoptosis of the transfected cells *in vitro*[8, 9]. In the present study, a HAdV5-based vector carrying the *PUMA* gene under the control of the cytomegalovirus (CMV) immediate-early promoter (HAdV5-PUMA) was tested *in vitro* in human and rat FLSs and evaluated for the

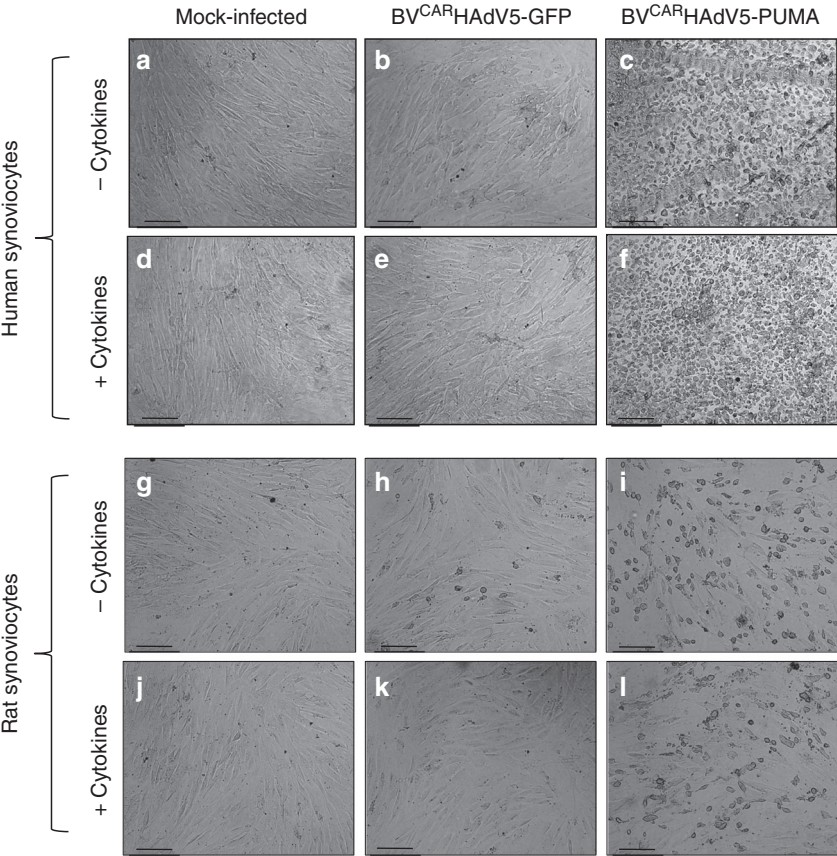

**Fig. 2** Cell death of human and rat FLS induced by BV$^{CAR}$HAdV5-PUMA. Cultures of human and rat FLS were non-treated **a–c,g–i** or treated **d–f,j–l** with proinflammatory cytokines, TNFα and IL-17. The FLS samples were mock-infected **a,d,g,j**, infected with BV$^{CAR}$HAdV5-GFP **b,e,h,k** or infected with BV$^{CAR}$HAdV5-PUMA **c,f,i,l**. Note the rounding-up of FLS after transduction with BV$^{CAR}$HAdV5-GFP in **c,f,i,l**. Scale bars, 20 μm

cytological effects due to PUMA expression and a HAdV5-GFP vector is used as control.

Both HAdV5 vectors were used at the same vector dose of 30 vp/cell and in complex with BV$^{CAR}$. No morphological changes were observed in both the mock-transduced and BV$^{CAR}$HAdV5-GFP-transduced FLS at 24 h post transduction (Fig. 2a,b,g,h). However, both human and rat FLS transduced with BV$^{CAR}$HAdV5-PUMA show massive cell death (Fig. 2c,i). To mimic the inflammatory conditions in arthritic joints, human and rat FLSs were pre-treated with proinflammatory cytokines TNFα and IL-17 before vector transduction (Fig. 2d,e,f,j,k,l). Similarly, massive cell death was observed in both types of FLS transduced with BV$^{CAR}$HAdV5-PUMA (Fig. 2f,l). This implied that cell death of both human and rat FLS could be induced by an HAdV5 vector expressing a proapoptotic gene such as *PUMA*, and achieved with the cooperation of a BV$^{CAR}$ vector. Since rat FLS are sensitive to PUMA-induced apoptosis, a rat *in vivo* model of arthritis could be evaluated with the BV$^{CAR}$HAdV5-PUMA vector.

For clinical relevance, we evaluated FLS derived from synovium explants of three RA patients, for their sensitivity to PUMA-induced cell death. BV$^{CAR}$HAdV5-PUMA transduction was carried out at a constant vector dose of 50 vp/cell, with or without pretreatment with proinflammatory cytokines, and the cell survival measured by the MTT assay over a period of 40 h post infection. The cell survival curves from the three clinical samples show similar profiles, with a rapid cell death at 16 h post infection, followed by a more gradual effect until 40 h (Fig. 3a–c). The percentage of surviving cells at 40 h post transduction was 20% or less in all three samples, and minor differences in the cell

survival curves observed in the presence or absence of cytokines (Fig. 3a–c). The results show that BV$^{CAR}$HAdV5-PUMA transduction of the FLS derived from the three clinical samples induced rapid cell death in all samples, with or without cytokines.

Protein analysis of the three clinical samples taken before BV$^{CAR}$HAdV5-PUMA transduction show low basal levels of PUMA protein (Fig. 3d). After transduction of the FLS by BV$^{CAR}$HAdV5-PUMA, high expression of PUMA protein was detected in all three samples (Fig. 3d and Supplementary Fig. 1). The results strongly implied the role of PUMA protein expression in the induction of FLS cell death.

To further confirm that the cell death of human FLS transduced with BV$^{CAR}$HAdV5-PUMA was due to apoptosis, an enzyme-linked immunosorbent assay (ELISA) was performed to measure the degree of chromatin fragmentation and nucleosome released in the cell cytoplasm. Non-transduced FLSs served as the control of background cytoplasmic nucleosomes, whereas FLSs transduced with BV$^{CAR}$HAdV5-GFP served as control for cytotoxic effects of HAdV5 vector transduction. For FLS samples transduced with the lowest vector dose of 10 vp/cell BV$^{CAR}$-HAdV5-PUMA, the amount of cytoplasmic nucleosomes was significantly higher compared to control samples (Fig. 3e), suggesting induction of FLS apoptosis. The level of cytoplasmic nucleosomes increased at higher vector doses, reaching a plateau at the relatively low dose of 20 vp/cell (Fig. 3e). A similar profile was obtained with FLS pre-treated with the cytokine TNFα (Fig. 3e). For FLS pre-treated with IL-17, individually or in combination with TNFα, the overall levels of cytoplasmic nucleosomes were lower, compared to those of non-treated or TNFα-treated FLS (Fig. 3e).

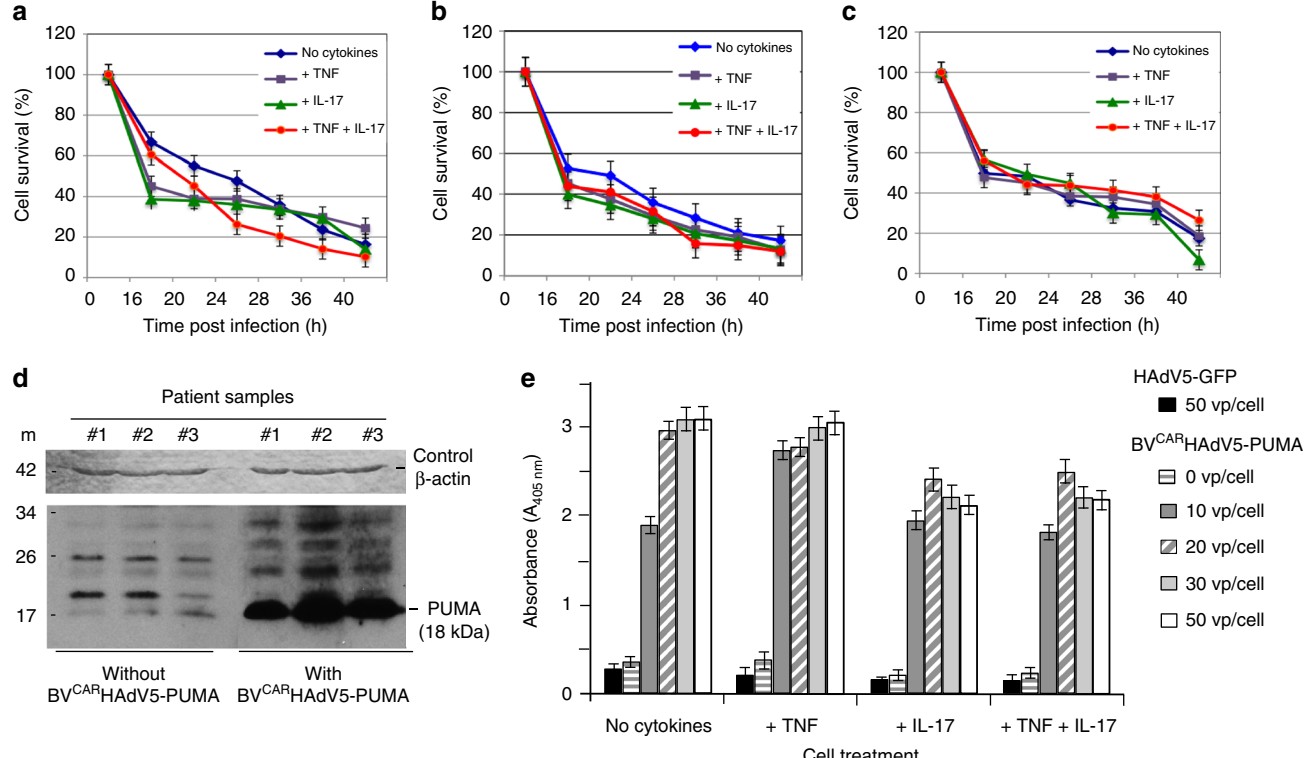

**Fig. 3** BV^CAR HAdV5-PUMA-induced apoptosis of FLS derived from clinical samples. FLS derived from synovium explants from three patients with RA **a–c**, nontreated or treated with cytokines TNFα and IL-17, were infected with BV^CAR HAdV5-PUMA at a constant adenoviral vector dose of 50 vp/cell. The cell survival was monitored over 40 h, using the MTT assay. The number of surviving cells was expressed as the percentage of control cells taken at time 0 of infection, which was attributed the 100 per cent value. The average from three independent experiments, each performed in triplicates is presented. The error bars indicate standard error of the mean. Statistical analysis using one-way ANOVA test, ***$P < 0.001$, **$P < 0.01$, *$P < 0.05$. **d** The expression of PUMA protein was analysed in FLS transduced or not with BV^CAR HAdV5-PUMA at 50 vp/cell. Cells were harvested at 40 h post transduction, lysed and whole-cell lysates probed for PUMA protein using SDS–polyacrylamide gel electrophoresis and western blot analysis. (*bottom*) Luminogram of blot reacted with anti-PUMA RabMAb, peroxidase-conjugated anti-rabbit IgG antibody and enhanced chemiluminescence Detection Kit. (*Top*) Loading control (stained blot). Blot was reacted with anti-β-actin MoMAb, peroxidase-conjugated anti-mouse IgG antibody, and staining reaction performed with $H_2O_2$ and 3,3'-Diaminobenzidine. **e** To quantitatively measure cell apoptosis, FLS untreated or treated with individual cytokines TNFα and IL-17, or a mixture of TNFα and IL-17, were incubated with increasing doses of BV^CAR HAdV5-PUMA, and harvested at 24 h post infection. DNA fragmentation and nucleosome release into the cytoplasm were determined using an immunological detection of histone-complexed DNA fragments. Lysates from mock-infected cells served as negative controls to evaluate the background level, i.e., the physiological nucleosome content of the cytoplasm. The results are expressed as the fold change over the background level, which was attributed the value of 1. The control consisted of FLS infected by BV^CAR HAdV5-GFP at the highest dose of 50 vp/cell. Results presented are from three independent experiments, each in triplicates, using the one-way ANOVA test, ***$P < 0.001$, **$P < 0.01$, *$P < 0.05$

**Effect of BV^CAR HAdV5-PUMA treatment in rat arthritis model**. We next explored the effect of BV^CAR HAdV5-PUMA injection into the synovial cavity of rats with adjuvant-induced arthritis (AIA). This is an animal model of oligo-articular arthritis in which the ankles are particularly affected, with the development of arthritis 8–10 days after adjuvant injection. Rats were given different vector preparations for a single intra-articular injection at day 14 post induction, i.e., at the active stage of the disease, as would be the case in a clinical setting. Thirty rats were divided into six groups (three control and three therapeutic) of five rats each. Rats of control groups were injected with (i) BV^CAR alone ($10^5$ plaque forming unit (PFU) per joint), (ii) BV^CAR-HAdV5-null ($10^5$ PFU BV^CAR + $10^9$ PFU HAdV5-null per joint) and (iii) HAdV5-PUMA alone ($10^9$ PFU per joint). Rats of the three therapeutic groups were administered BV^CAR ($10^5$ PFU per joint) complexed with HAdV5-PUMA at three different doses ($10^7$, $10^8$ and $10^9$ PFU per joint, respectively) and monitored for 4 days.

Rats which received the control vectors BV^CAR or HAdV5-PUMA alone, or the BV^CAR HAdV5-null, showed no therapeutic effect on the expected course of arthritis (Fig. 4). However, rats treated with BV^CAR HAdV5-PUMA showed significant reduction of the ankle circumference in a vector dose-dependent manner (Fig. 4a). Likewise, there was a decrease in the ankle articular index score, although this was only significant for the group treated with the highest HAdV5-PUMA vector dose (Fig. 4b). The body weight of rats which received BV^CAR HAdV5-PUMA was superior to the values of control animals for all the vector doses tested (Fig. 4c). Collectively, the intra-articular administration of BV^CAR HAdV5-PUMA after arthritis onset showed beneficial effect as early as 4 days after injection, with significant reduction of the ankle circumference indicating decreased joint inflammation.

Peri-articular bone damage was assessed using micro-computed tomography (μ-CT). The three-dimensional images allow an overall evaluation of bone loss in the ankles, while the parasagittal slices would provide more refined details. When compared to the healthy ankle (Fig. 5a), substantial bone damage was observed in the joints injected with BV^CAR alone (Fig. 5b), and extensive bone loss was also visible in the joints treated with

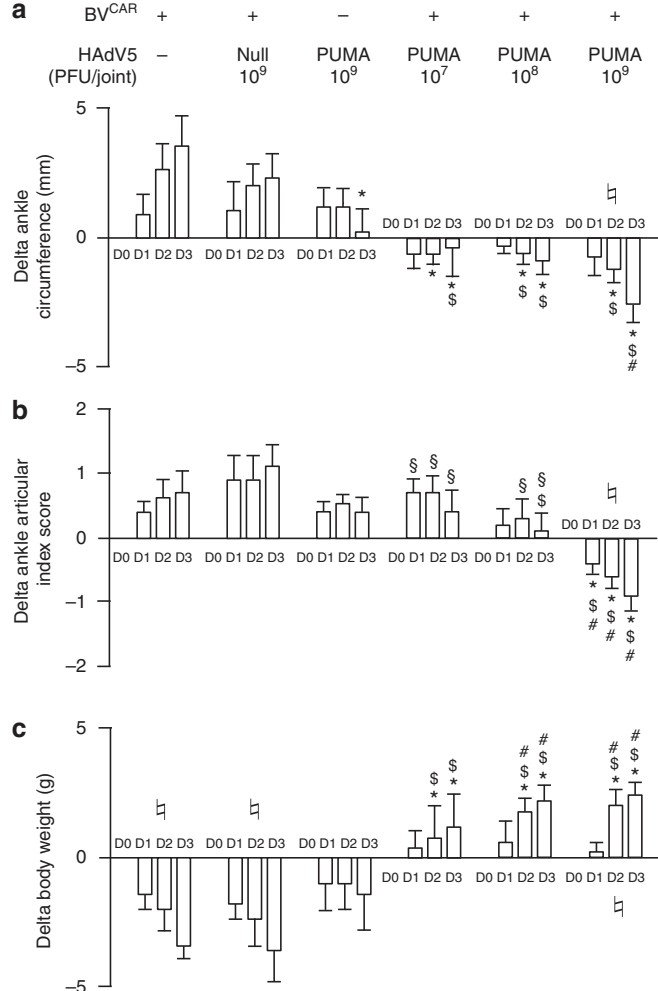

**Fig. 4** BV$^{CAR}$HAdV5-PUMA-mediated reduction of joint inflammation. Rats develop arthritis 8–10 days after adjuvant injection. After the onset of arthritis, at day 14 post injection, vectors (10 μl) were delivered intra-articularly in each ankle. Thirty rats were enrolled and divided into six groups (five rats per group). Three control groups consisted of BV$^{CAR}$ alone (10$^5$ PFU per joint), BV$^{CAR}$ (10$^5$ PFU per joint) complexed with an empty adenoviral vector (HAdV5-null; 10$^9$ PFU per joint), and HAdV5-PUMA alone (10$^9$ PFU per joint). The three therapeutic groups consisted of BV$^{CAR}$ (10$^5$ PFU per joint) complexed with HAdV5-PUMA at increasing concentrations (10$^7$, 10$^8$ and 10$^9$ PFU per joint). Values represented in the bar graphs are the means ± s.e.m. Differences (Delta) in the biological parameters are defined by the values obtained on the day of follow-up minus the values obtained on the day of intra-articular injection. **a** Differences in ankle circumferences. Values of ankle circumferences were obtained from perpendicular caliper measurements of ankle diameter, using a geometric formula. **b** Differences in ankle articular index scores. **c** Differences in rat body weights. D day after intraarticular injection of vector; *$P < 0.05$ vs BV$^{CAR}$ alone; $^\$P < 0.05$ vs BV$^{CAR}$HAdV5-null; #$P < 0.05$ vs HAdV5-PUMA alone; $^\$P < 0.05$ vs BV$^{CAR}$HAdV5-PUMA (10$^9$ PFU per joint); $^\natural P < 0.05$ for ANOVA one-way test

HAdV5-PUMA alone (Fig. 5c). In contrast, only minor morphological changes were observed in joints administered with BV$^{CAR}$HAdV5-PUMA (Fig. 5d,e), as compared to ankle joints from healthy rats (Fig. 5c). The only bone loss marker detectable in BV$^{CAR}$HAdV5-PUMA-treated joints was a modification in porosity, characteristic of the early phase of arthritis before the administration of any treatment. Further

microarchitecture alterations consecutive to extended inflammation, was not observed in the BV$^{CAR}$HAdV5-PUMA-treated joints.

Synovium changes were evaluated by light microscopy of fresh-frozen sections of the joints of euthanized animals, and conventional staining with haematoxylin and eosin (H&E). In comparison to the control joints treated with BV$^{CAR}$ or HAdV5-PUMA alone, the infiltrate was significantly reduced in the BV$^{CAR}$HAdV5-PUMA-treated joints (Fig. 5f). This effect was vector dose-dependent, with a lower number of infiltrating cells in ankle joints treated with 10$^9$ PFU compared to 10$^7$ PFU of BV$^{CAR}$HAdV5-PUMA (Figs 5f and 6).

**Long-term effect of BV$^{CAR}$HAdV5-PUMA treatment in rat model.** To assess the long-term therapeutic effect, an animal experiment was performed with rats treated with BV$^{CAR}$HAdV5-PUMA and HAdV5-PUMA (10$^9$ PFU per joint), and the study period extended up to 21 days post injection. Following restrictions by the animal ethic committee, the negative control groups (BV$^{CAR}$ and BV$^{CAR}$HAdV5-null) were not included in this long-term study due to the high level of arthritis severity.

The results show that the articular index decreased from day 2 to day 21 post-intra-articular injection (Fig. 7a). A similar pattern was observed for the mobility index (Fig. 7b) and ankle circumference (Fig. 7c). In comparison to the rats which received HAdV5-PUMA, the mobility of the BV$^{CAR}$HAdV5-PUMA-treated animals improved progressively from day 8 to day 21 and their body weight improved rapidly from day 3, remaining significantly higher until the end of the study (Fig. 7d). All the parameters measured confirmed the therapeutic benefits of BV$^{CAR}$HAdV5-PUMA treatment with improvement of the joint function, and absence or minimal joint and bone damage, as observed in histopathology and micro-computed tomography. Control animals administered with HAdV5-PUMA alone showed no detectable beneficial effect.

Histo-morphometry analysis was performed on the treated joints, to study the protective effects of BV$^{CAR}$HAdV5-PUMA treatment on the bone and cartilage. The tartrate-resistant acid phosphatase+ (TRAP+) osteoclast staining was reduced at several sites of the ankle area, including distal tibia, talus, calcaneus and navicular bones (Fig. 7e). Two areas remained TRAP+ in distal tibia and calcaneus, corresponding to the remodeling of growth plates, which are active in rats at this age. Elsewhere, pathologic osteoclastogenesis was abolished in the BV$^{CAR}$HAdV5-PUMA group. Mineralized bone was also protected, in the same sites as described with less TRAP+ osteoclasts (Fig. 7f). More mineralized trabeculae and less non-mineralized areas were observed in the bones of the BV$^{CAR}$HAdV5-PUMA group in comparison with the HAdV5-PUMA group. Finally, the regularity and the thickness of the cartilage were preserved in the BV$^{CAR}$HAdV5-PUMA group, especially at the talo-tibial joint site (Fig. 7g).

**Vector dessemination and persistence.** To assess if the HAdV5-PUMA and BV$^{CAR}$HAdV5-PUMA vectors remain localized within or escaped from the injected joints, we performed PCR assays on total DNA extracted from liver samples and sera from rats killed at day 4 and day 21 after treatment. No amplification of either adenovirus or baculovirus genome were obtained from the liver and serum samples. These results implied that there was no detectable dissemination of either vectors from the injected joints.

To address the persistence of the vectors in the injected joints, synovial tissues from animals injected with HAdV5-PUMA and BV$^{CAR}$HAdV5-PUMA and sacrificed at day 21 were analysed. Synovial tissues from four individual rats from each group were chosen at random and the total DNA extracted. PCR assays,

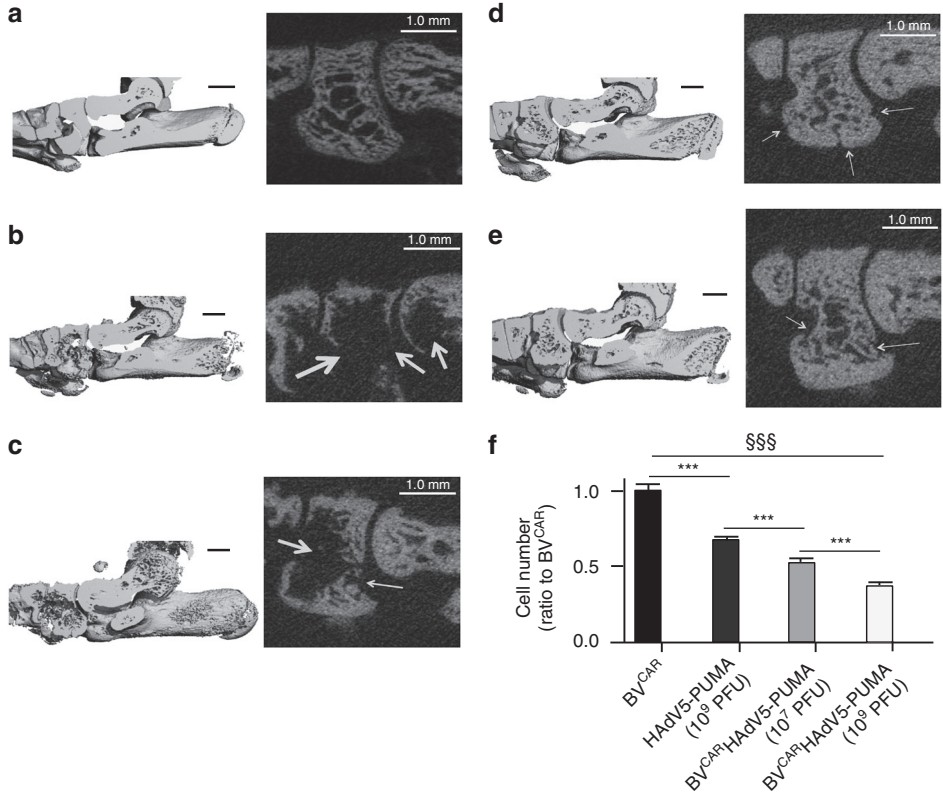

**Fig. 5** Analysis of rat ankle tissues. Ankle tissue imaging by micro-computed tomography (μ-CT) was performed on rat ankle joints from **a** a healthy rat, rats injected with **b** BV$^{CAR}$ alone, **c** HAdV5-PUMA alone and animals injected with BV$^{CAR}$ HAdV5-PUMA at vector doses of **d** $10^7$ PFU per joint, and **e** $10^9$ PFU per joint respectively. (*left*) Parasagittal three-dimensional views of the whole rat ankles. (*right*) The enlarged images of the μ-CT parasagittal slices showing, from *left* to *right*, the intermediate cuneiform, navicular and distal talus. The white arrows point at local bone erosions. Scale bars, 1 mm. **f** Histological analysis and quantification of infiltrates was also performed. The bar graph shows the number of infiltrating cells, expressed as the ratio to their number in BV$^{CAR}$ control samples (mean ± s.e.m.; data from 30 slides per group). ANOVA: analysis of variance; §§§: ANOVA, $P < 0.001$; ***$P < 0.0001$ by Newman–Keuls *post-hoc* tests. Note that (i) BV$^{CAR}$HAdV5-PUMA strongly reduced the density of inflammatory cells in the joints, compared to control BV$^{CAR}$ or HAdV5-PUMA; (ii) BV$^{CAR}$HAdV5-PUMA was more effective at $10^9$ PFU, compared to $10^7$ PFU, demonstrating a dose-dependent effect

performed using primers recognizing the hexon gene of HAdV5, amplified the expected hexon band of 465 bp in all the samples (Fig. 8a; arrowhead). However, no specific baculovirus band of 1.1 kbp, corresponding to a sequence within the baculovirus CAR gene, was detected in any of the synovium samples (Fig. 8b). These results implied the persistence of the HAdV5-PUMA genome in the synovial tissues at least until day 21 post treatment, but not that of the BV$^{CAR}$ genome.

**Immune response to gene delivery vectors**. To evaluate if there was immune response to the viral vectors, sera from rats treated for 4 days and 21 days with BV$^{CAR}$, HAdV5-PUMA or BV$^{CAR}$HAdV5-PUMA, were analysed by ELISA for the presence of antibodies to HAdV5 and BV$^{CAR}$. The results show that antibodies against both HAdV5 and BV$^{CAR}$ were either absent or not detectable from the day-4 sera (Fig. 8c,d). However, from the day-21 sera, anti-HAdV5 antibodies were detected in 2/4 animals of the HAdV5-PUMA group, and in 3/4 animals of the BV$^{CAR}$HAdV5-PUMA group (Fig. 8c). Low levels of anti-BV antibodies were detected in the sera of 2/4 animals of the BV$^{CAR}$HAdV5-PUMA group (Fig. 8d). Although no viral genomes were detectable in the sera and tissues, the presence of antibodies against the viral vectors in certain individual animals suggested that the immune response observed could be due some vector leakage from the treated joints or induction of Ig by activated B cells inside the synovium[15].

## Discussion

The aim of this study is to show the feasibility of controlling arthritis severity by an apoptosis-inducing gene therapy strategy targeting the FLSs in arthritic joints. HAdV5 vectors have been widely used as gene transfer vectors for cancer therapy. However, the use of HAdV5 vectors for arthritis treatment are limited as FLSs are poorly transduced and attempts to improve the gene transfer efficacy by genetic modification of the vector were not satisfactory[13, 16]. The innovation in the present study is the development of a strategy for efficient HAdV5 gene transfer to FLS coupled to a potent apoptosis-inducing gene for the treatment of chronic synovitis.

Our strategy consists of an HAdV5 vector carrying a proapoptotic gene *PUMA*, complexed on recombinant baculovirus (BV) displaying CAR on the baculoviral envelope (BV$^{CAR}$)[14]. The BV$^{CAR}$HAdV5-PUMA complex exploits the capacity of the BVs to enter FLSs, resulting in efficient HAdV5 transduction of FLS *in vitro* and *in vivo*. Both the HAdV5 and BVs are non-integrative vectors and are considered biologically safe. BVs have a biosafety profile due to their incapacity to replicate in mammalian cells[17, 18].

The functionality and efficiency of our strategy for *PUMA* gene delivery by an HAdV5-PUMA vector was demonstrated *in vitro* as well as *in vivo* in a rat AIA model. Gene transfer of *PUMA* into FLSs by BV$^{CAR}$HAdV5-PUMA, results in a rapid and massive cell death of FLS *in vitro*. In the context of chronic joint

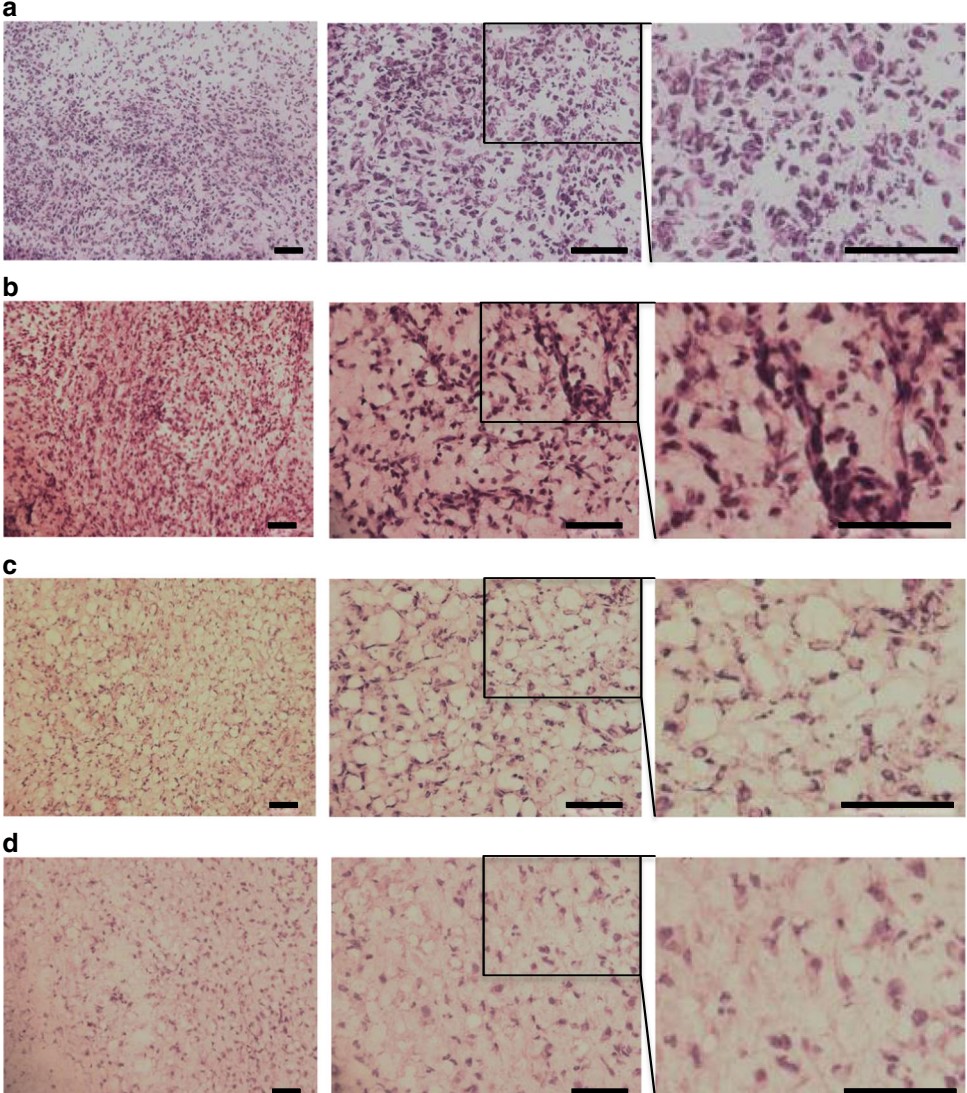

**Fig. 6** Histological analysis of ankle joints. Fresh-frozen sections (10 μm) of joints treated respectively with: **a**, BV^CAR alone; **b**, HAdV5-PUMA alone ($10^9$ PFU); **c**, BV^CAR HAdV5-PUMA ($10^7$ PFU); **d**, BV^CAR HAdV5-PUMA ($10^9$ PFU); were stained with haematoxylin and eosin staining, and compared for the presence of synovial tissue infiltrates. The representative photograph from each group is provided at various magnifications (×200, *left*; ×400, *middle*; and ×800, *right*). The righmost panels are enlargements of the rectangular area delineated in the middle panels. Scale bar, 200 μm

inflammation, the persistence of the disease results in part from the reduced sensitivity of FLS to death signals[19]. We also show that the FLS death cell induced by the HAdV5-PUMA vector is not inhibited in the presence of inflammatory cytokines. Using the rat AIA model, a single intra-articular injection of BV^CAR HAdV5-PUMA decreases significantly joint inflammation and improves joint function, resulting in minimal joint damage and bone loss. The two videos (Supplementary Movies 1 and 2) show that the treatment not only restores the joint functions and ambulation but also the mannerisms of the treated animal. The HAdV5-PUMA vector remains detectable in the injected joints 21 days post treatment, suggesting that the PUMA therapeutic effect can prolong in the joints beyond the observation period. The induction of antibodies against the adenovirus vector and its effect on the efficacy of intra-articulation treatment in humans is unknown. However, the possible adverse effects should be limited if a single therapeutic administration is proposed. Alternatively, chimeric adenovirus vectors with capsid proteins from other serotypes could replace the conventional HAdV5 vector to

escape the circulating and local adenovirus neutralizing antibodies.

The intra-articular treatment options today are limited to intra-articular steroids with only transient anti-inflammatory effects. Other options such as radiochemical synovectomy with radio-active isotopes or osmic acid are not easily or not at all available. Our results show that local intra-articular treatment with the administration of a PUMA-expressing vector represents an efficient strategy for the control of abnormal synoviocyte proliferation. In addition to the local protective effect, the control of local inflammation appears to have a systemic effect. Nevertheless, this apoptosis-based therapeutic strategy has potential as an alternative to surgical or radiochemical synovectomy in a long list of arthritis conditions including RA, mono- or oligo-articular arthritis which includes juvenile oligo articular arthritis, inflammatory osteoarthritis, or the orphan disease pigmented-villo-nodular synovitis. A first proof of concept trial to treat this cancer-like disease by inducing targeted cell death may help in preventing the high rate of recurrences.

## Methods

**Ethical approval.** The use of human patient biological samples (FLS) in this study was approved by the Institutional Review Board of the Lyon University Hospital and the French Ministry of Education and Research (Authorization #AC-2010-1164). Informed consent was obtained from the patients. Animal experiments were performed in full compliance with the Ethical Committee Guidelines and Regulations for Animal Protection of the Loire in accordance with the legislation of the European Community, and received approval from the Ethical Committee for Animal Experiments of the University Jean Monnet in Saint-Etienne (Authorization #CU-14N1402).

**Cells.** Human embryonic kidney cells HEK293 cells were obtained from the American Type Culture Collection (ATCC, Manassas, VA), and maintained in Dulbecco's modified Eagle's medium (DMEM; Life Technologies) supplemented with 10% fetal bovine serum (FBS; Life Technologies), penicillin (100 U ml$^{-1}$), and streptomycin (100 mg ml$^{-1}$) at 37 °C, 5% $CO_2$. *Spodoptera frugiperda* cells

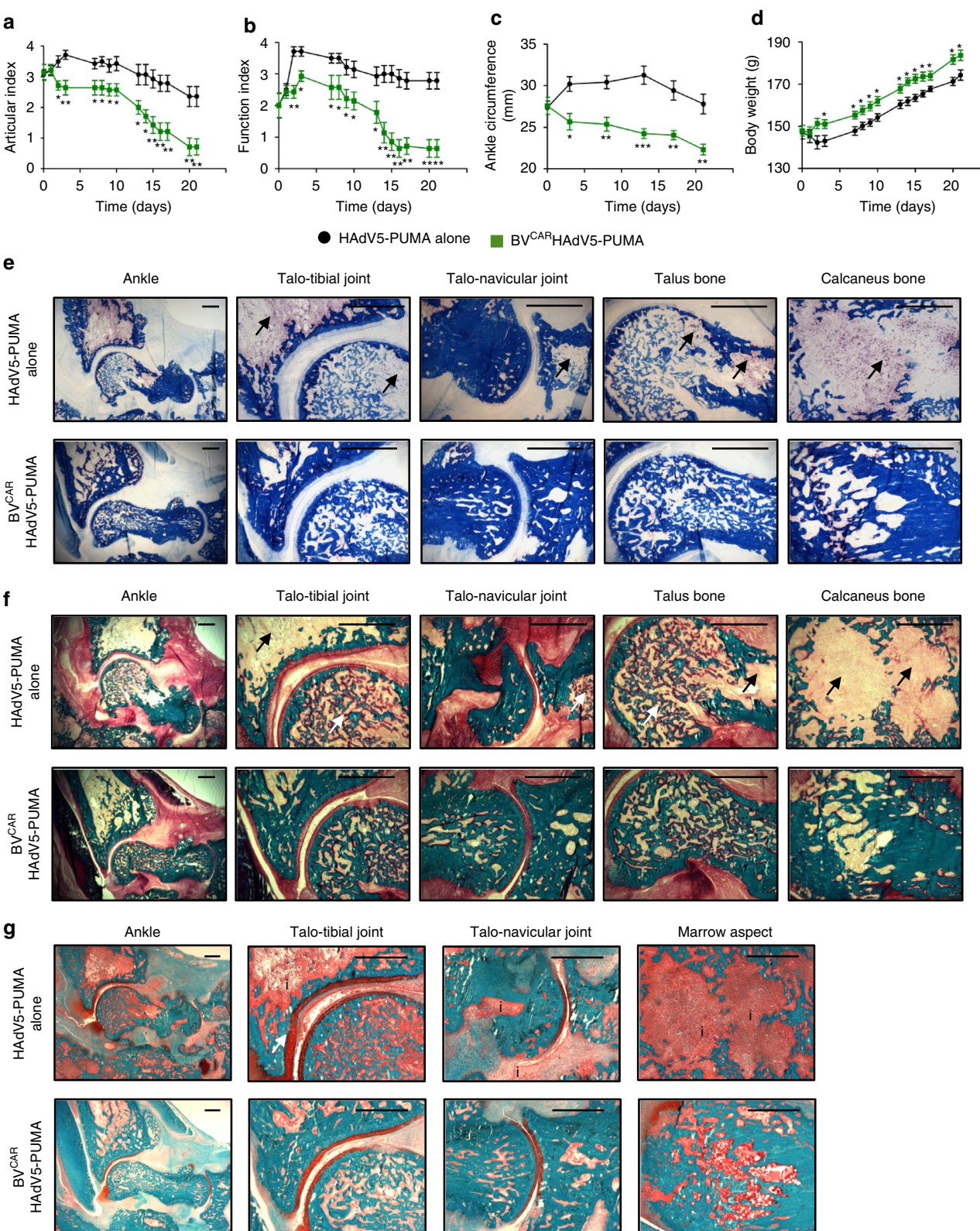

(purchased from Life Technologies) were maintained at 28 °C in Grace insect medium supplemented with 10% FBS and penicillin and streptomycin (Life Technologies). FLS were obtained from synovial tissue of RA patients undergoing joint surgery who fulfilled the ACR/EULAR criteria for RA[20]. Human and rat FLS were isolated by enzyme digestion and cultured in DMEM supplemented with 10% FBS and antibiotics and used between passages 4 and 9, as previously described[13]. All cell lines were routinely checked for mycoplasma every 6 months.

**Antibodies and inflammatory cytokines.** Mouse monoclonal antibody (MAb) against CAR (clone E1.1)[21] obtained from Silvio Hemmi (University of Zürich, Switzerland) was used at dilution 1:50. MAb against the baculoviral glycoprotein Gp64 (clone AcV1) used at dilution 1:50, were purchased from Santa Cruz Biotechnology. Rabbit MAb against PUMA (clone EP512Y) used at dilution 1:1000 (Epitomics Inc.) was purchased from Abcam (Cambridge, UK). MAb against β-actin-peroxidase (clone AC-15) from Sigma-Aldrich was used at dilution 1: 10,000. TNFα was used at 1 ng ml$^{-1}$ (Biosource, Camarillo, CA), IL-17A at 50 ng ml$^{-1}$ (R&D Systems, Minneapolis, MN).

**Adenovirus and baculovirus vectors.** The replication-defective (E1-deleted) HAdV5-based vectors expressing the green fluorescent protein (HAdV5-GFP) and the human PUMA−β protein (HAdV5-PUMA) under the control of the CMV immediate-early promoter were propagated in HEK293 cells. The HAdV5-null vector, harboring no transgene in the E1-deleted region of its genome, was purchased from GeneCust (Dudelange, Luxembourg). The vector stocks were prepared and purified by CsCl gradient ultracentrifugation according to conventional methods[14, 22, 23]. The recombinant baculovirus (BV) vector expressing CAR (BV$^{CAR}$) was derived from the *Autographa californica* Multiple Nucleopolyhedrosis virus (AcMNPV) by inserting the human CAR-encoding sequence into the *Nhe* I and *Kpn* I cloning sites of pBlueBac (Life Technologies) under the control of the polyhedrin promoter[14]. The BV vector expressing GFP under the CMV promoter (BV-GFP) was a kind gift of Norman J. Maitland (University of York, UK). BV vectors were propagated by infection of Sf9 cells at a multiplicity of infection (MOI) of 1–5[14]. The infected cell culture media were harvested at 48 h post-infection and clarified by centrifugation for 10 min at 2400 r.p.m. Concentrated stocks of BV vectors were prepared by ultracentrifugation of the infected cell supernatant through a 20% sucrose cushion at 30,000 r.p.m. for 1 h at 4 °C. The viral pellet was then resuspended in sterile phosphate-buffered saline (PBS) with gentle shaking overnight at 4 °C, and used for transduction assays.

**Cell transduction assays.** The infectious titre of a virus is defined as the number of infectious virions, determined by the plaque assay method and expressed as (PFU) per ml. In our transduction assays, cells were prepared in 24-well plates containing 10$^5$ cells per well. Complexes of adenovirus and baculovirus vectors in the ratio 5:100 PFU respectively, were prepared by preincubation of each virus in a total volume of 100 µl of cell culture media for 1 h at 37 °C. The complexes were then added to the cells and incubated for another hour at 37 °C, after which 200 µl prewarmed media was added to each well and incubated at 37 °C.

**Light microscopy.** Twenty-four hours post transduction, transduced cells were observed directly from the tissue culture plate using a Zeiss Axiovert inverted microscope (Zeiss). Images were taken using an Axiovert digital camera and analysed using an Axio Vision program (Zeiss).

**Electron microscopy and immuno-electron microscopy.** Vector samples were diluted in 0.14 M NaCl, 0.05 M Tris-HCl buffer pH 8.2 (Tris-buffered saline, TBS), and adsorbed onto carbon-coated Formvar membranes on grids, and incubated with primary antibody (monoclonal anti-CAR or anti-gp64 antibody) at a dilution of 1:50 in TBS for 1 h at room temperature (RT). After rinsing with TBS, the grids were postincubated with 20-nm colloidal gold-tagged goat anti-mouse immunoglobulin G antibody (British Biocell International Ltd., Cardiff, UK; diluted to

1:50 in TBS) for 30 min at RT. After rinsing with TBS, the specimens were negatively stained with 1% uranyl acetate in H$_2$O for 1 min at RT, rinsed again with TBS, and examined under a JEM1400 JEOL electron microscope equipped with an Orius-Gatan digitalized camera (Gatan, Grandchamp, France).

**Cell toxicity and apoptosis assays.** Cells grown in 96-well flat-bottom plates were transduced with our different vectors and analysed at 36 h post-transduction. The cell culture media was then removed, and 30 µl of MTT solution (7.5 mg ml$^{-1}$ thiazolyl blue tetrazolium bromide (Sigma-Aldrich) in phosphate buffered saline) was added per well, followed by incubation at 37 °C for 4 h. The MTT solution was then removed and 100 µl of DMSO (dimethylsulfoxide, Sigma-Aldrich) was added to each well. The optical density of the supernatants in the 96-well plates was read at 570 nm.

The cell apoptosis assay was carried out using a Cell Death Detection ELISA kit (Roche Diagnostics). Briefly, aliquots of FLS (10$^5$ cells) were harvested at different times post-infection and analysed by immunodetection to measure the degree of release of histone-complexed DNA fragments from each sample, according to the manufacturer's recommendations.

**Detection of adenovirus and baculovirus DNA.** Total nucleic acids were extracted from rat sera or synovium, using the NucleoSpin Tissue kit from Macherey Nagel, following the manufacturer's protocol. The presence of adenovirus and baculovirus DNA genomes in the rat sera and synovium were then evaluated by PCR using primers specific for each virus. For the HAdV5, the forward primer 5′-GCTGTATTTGCCCGAC-3′ and reverse primer 5′-CATGGCCTCAAGCGTG-3′, amplified a fragment of 465 nt of the hexon gene. For the baculovirus, the forward primer 5′-GCTGTATTTGCCCGAC-3′ and reverse primer 5′-CATGGCCTCAAGCGTG-3′, amplified a fragment of 1095 nt of the baculovirus-CAR gene.

**Detection of antibodies against adenovirus and baculovirus.** Purified HAdV5-PUMA in PBS were dissociated by incubation with 0.05% deoxycholate at 56 °C for 2 mins, while BV$^{CAR}$ virions in PBS were dissociated by a treatment of four cycles of freeze-thawing. Microtitre plates were coated with 100 µl of the viral proteins (1 µg ml$^{-1}$) diluted in PBS and left overnight at 4 °C in a moisture chamber. The coated wells were washed four times with PBS and incubated with 200 µl blocking solution (2% bovine serum albumin in TBS) for 1 h at RT. After washing, 100 µl of each sera was added to the wells and incubated for 1 h at RT. Plates were then washed and incubated with anti-rat-horseradish peroxidase diluted to 1: 15,000 in blocking solution for 1 h at RT. After washing, 100 µl of OPD (o-phenylenediamine, Sigma-Aldrich) substrate was added, and the reaction was blocked by addition of 1 N HCl. OD was measured at 450 nm using an ELISA plate reader (Wallac 1420 Multilabel Counter, PerkinElmer).

**Polyacrylamide gel electrophoresis and western blot analysis.** Polyacrylamide gel electrophoresis of SDS-denatured protein samples, and immunoblotting analysis have been described in detail in previous studies[14, 22–25]. Briefly, proteins were electrophoresed in SDS-denaturing, 10%-polyacrylamide gel, along with prestained protein markers (PageRuler$^{TM}$ prestained protein ladder; Fermentas Inc., Hanover, MD), and electrically transferred to nitrocellulose membrane (Hybond$^{TM}$-C-extra; GE Healthcare Bio-Sciences). Blots were blocked in 5% skimmed milk in TBS containing 0.05% Tween-20 (TBS-T), rinsed in TBS-T, then successively incubated with primary antibody anti-PUMA protein (RabMAb; working dilution 1:1000) and peroxidase-conjugated secondary antibody (goat anti-rabbit IgG antibody; Sigma, St Louis, MO; working dilution 1:10,000), followed by enhanced chemiluminescence detection (Pierce Chemicals, Thermo Fisher Scientific Inc. IL).

**Rat adjuvant-induced arthritis (AIA) model.** Female Lewis rats weighing ~100 g (Janvier Laboratories, Saint-Germain-sur-l'Arbresle, France), were injected subcutaneously at the base of the tail with 300 µl (5 mg ml$^{-1}$) of lyophilized

**Fig. 7** Long term *in vivo* and *in vitro* analysis of PUMA gene transfer. Rat ankle joints were injected with HAdV5-PUMA alone, or BV$^{CAR}$HAdV5-PUMA at a vector dose of 10$^9$ HAdV5-PUMA per joint, after the onset of arthritis at day 14. The parameters monitored over a period of 21 days were the **a** ankle articular index, **b** loss of function index, **c** ankle circumferences and **d** body weights. Histology was performed on undecalcified samples with various staining. Tartrate-resistant acid phosphatase (TRAP) staining provided red colour in multinucleated osteoclasts on the bone counter-stained with Anilin blue **e**. Compared to HAdV5-PUMA group, TRAP+ osteoclast staining with the TRAP was reduced at several sites of the ankle area, including distal tibia, talus, calcaneus and navicular bones in the HAdV5-PUMA group. Arrows showed important TRAP+ osteoclast concentration in the bones. In the same sites, the bone matrix was stained with Goldner trichrome staining **f**. Mineralized matrix was stained in green, while non-mineralized matrix was stained in red and the marrow in yellow. Arrows showed loss of bone mineral. More mineralized trabeculae and less non-mineralized areas were observed in the bones of the BV$^{CAR}$HAdV5-PUMA group in comparison with the HAdV5-PUMA group **f**. Then, Safranin O–Fast green staining was performed with bone tissue staining in turquoise and cartilage staining in red **g**. Compared to HAdV5-PUMA, cartilage thickness and its regularity were preserved in the BV$^{CAR}$HAdV5-PUMA group. *Arrows* indicated the loss of mineral in the bones. Inflammatory infiltrate (indicated by the 'i') was importantly diminished in the HAdV5-PUMA group and the bone cavity conserved its histological integrity. Scale bar, 500 µm

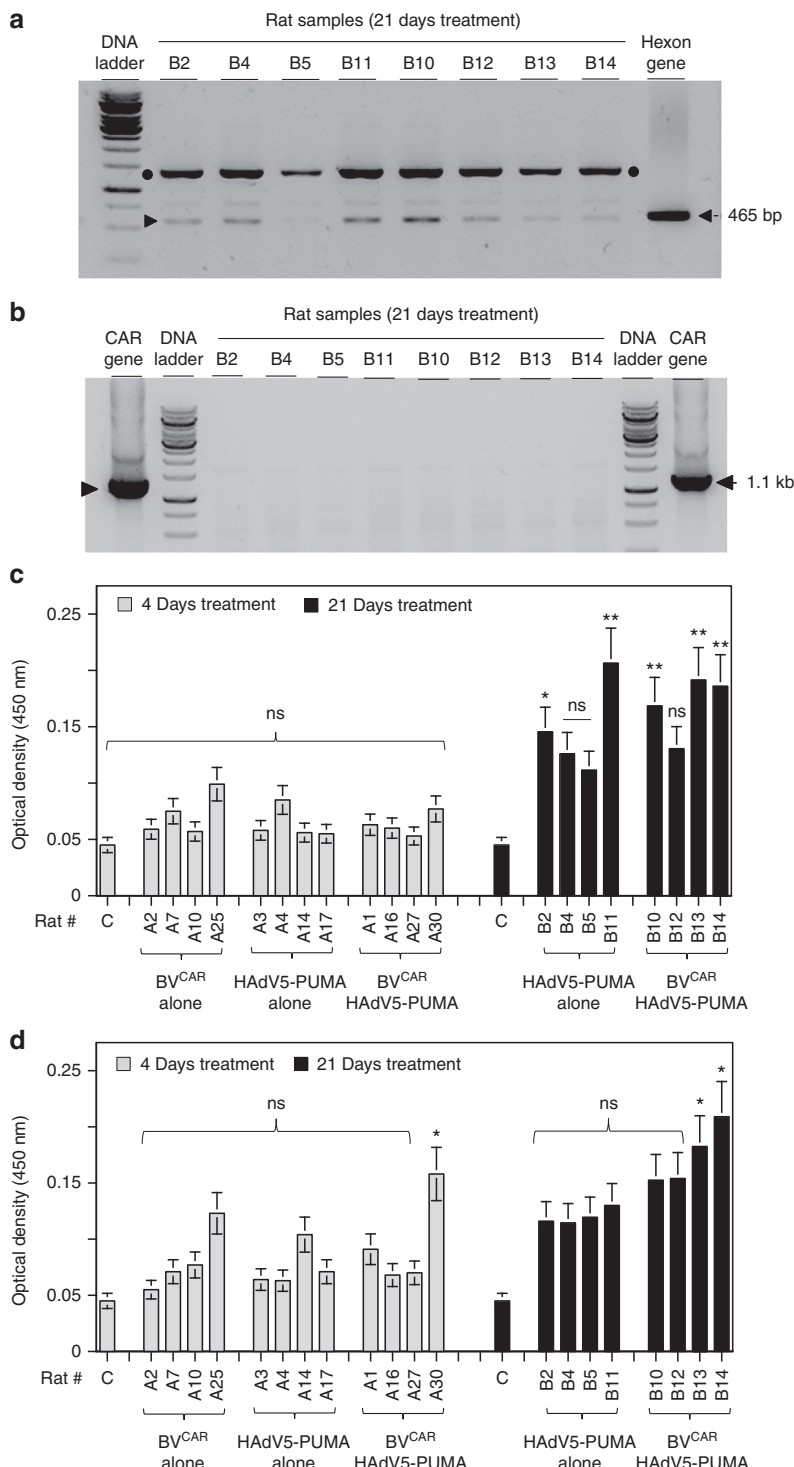

**Fig. 8** Analysis of vector persistence and immune response to vectors. **a** The HAdV5-PUMA and BV^CAR DNA genomes in synovial tissues were extracted from synovial tissues samples of rats killed at day 21 after intra-articular administration of HAdV5-PUMA alone (B2, B4, B5 and B11) or BV^CAR HAdV5-PUMA (B10, B12, B13 and B14), and used for PCR analysis to detect the viral genomes. **a** The 465-bp amplified fragment corresponding to the HAdV5 hexon gene (**a**, *arrowhead*) was found in all the samples and in the positive control. Note the occurrence of a non-specific amplified fragment of 1500 bp (*black dots*). **b**, For the BV^CAR genome, no specific band of 1.1 kbp, corresponding to the baculovirus CAR gene was detected in any of the samples, but was present in the positive control. **b** To evaluate the immune response of the treated animals to BV^CAR and HAdV5-PUMA vectors, the sera of the animals treated for 4 days and 21 days with HAdV5-PUMA alone or with BV^CAR HAdV5-PUMA were analysed by ELISA for the presence of antibodies against adenovirus and baculovirus. A single serum dilution (1:10) was used for all samples. No significant adenovirus or baculovirus antibodies were detected in the sera of the animals treated for 4 days (*grey bars*, **c**,**d**). In the animal group treated for 21-days, adenovirus antibodies were detected in the sera of 5/8 animals (*black bars*, **c**) and baculovirus antibodies detected in the sera of 2/8 animals (*black bars*, **d**). Similar results were obtained in an independent experiment

*Mycobacterium butyricum* (Difco Laboratories, Detroit, MI), as previously described[26]. In this model, the first signs of joint inflammation and pain appear after day-8 of induction and become maximal during days 14–18[26].

The various vectors were injected (intra-articular, 50 µl per ankle) 14 days after arthritis induction, defined as day 0. The first animal experiment was designed to assess short-term efficacy and to determine the best dose of vector. Thirty rats were divided into six groups (*n* = 5 per group). Three control groups were recombinant baculovirus alone (BV$^{CAR}$; $10^5$ PFU per joint), adenovirus vector harbouring the *PUMA* gene alone (HAdV5-PUMA; $10^9$ PFU per joint), and BV$^{CAR}$ ($10^5$ PFU per joint) in combination with an empty adenoviral vector (HAdV5-null; $10^9$ PFU per joint). The three therapeutic groups associated BV$^{CAR}$ ($10^5$ PFU per joint) with HAdV5-PUMA at three concentrations ($10^7$, $10^8$ and $10^9$ PFU per joint, respectively). In this experiment, the animals were euthanized on day 4 post-intra-articular injection.

The second animal experiment assessed the long-term efficacy of BV$^{CAR}$HAdV5-PUMA during 21 days compared to HAdV5-PUMA alone (both $10^9$ PFU per joint; *n* = 7 per group). The clinical parameters monitored during these experiments, including articular index, mobility index, ankle circumference and body weight. Articular index scores were recorded for each hind joint by a consistent observer blinded to the treatment regimen and then averaged for each animal. Scoring was performed on a 0–4 scale where 0 = no swelling or erythema, 1 = slight swelling and/or erythema, 2 = low-to-moderate oedema, 3 = pronounced oedema with limited joint usage and 4 = excess oedema with joint rigidity. Mobility index were also recorded in the same way and provided limb function with a scale from 0 to 4 where 0 = regular mobility, 1 = gait irregularities, intermittent lameness, 2 = continuous lameness, with use of the forefoot only, 3 = continuous lameness, use of the top of the paw by bending the metatarsals, and 4 = loss of function, no use of the paw. Ankle circumferences were evaluated as the perimeter calculated after antero-posterior and latero-lateral ankle measurement with digital caliper.

**X-ray micro-computed tomography and histological analysis.** Four days after intraarticular injection, rats were killed, and deskinned right ankles were collected, embedded in Neg50 (Thermo Scientific, Waltham, MA) and frozen in liquid nitrogen. Frozen right ankles were scanned by µ-CT (viva-CT40, Scanco, Brüti-sellen, Switzerland). Three-dimensional reconstructions were segmented using the following parameters: sigma, 2.8; support, 2; threshold, 289. After µ-CT acquisition, fresh-frozen sections of the joints (10 µm in thickness) from the different groups were stained with H&E, and examined for the presence of infiltrates[26].

**Histology on undecalcified samples.** After µ-CT assessment, dehydrated samples were embedded in methylmetacrylate resin and 9 µm slices were obtained. Analysis included TRAP staining for osteoclasts as multinucleated TRAP+cells on the bone counter-stained with Anilin blue. Goldner trichrome stained non-mineralized, osteoid tissue in red with Fuchsine, and mineralized bone in green with Light green. Safranin.O.–Fast green staining was used to analyse cartilage integrity.

**Statistical analyses.** For *in vitro* and *ex vivo* cellular analyses, data were presented as the mean of triplicate experiments (*m* ± s.e.m.), and were representative of results obtained from three independent experiments that produced similar results. Statistical analyses were performed using the Mann–Whitney test. For *in vivo* experiments in rats, average of ankle joints per animal was treated independently for statistical purposes. Differences between groups were calculated using analysis of variance (ANOVA) two-way tests with Bonferonni *post-hoc* tests. Comparison inside each group was performed using ANOVA one-way tests. *P* < 0.05 was considered as significant.

**Data availability**. The authors declare that the data supporting the findings of this study are available from the authors upon reasonable request.

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

## Acknowledgements

The work in the UCBL-INRA UMR-754 in Lyon was financed in part by the Contrat d'Interface INSERM-Hospices Civils de Lyon (CIF-2008-2013) and the French Foundation for Cystic Fibrosis (contract VLM-RF2013-0500796), and by the IHU prometteur OPERA. S.S.H. is a scientist of the French Institute of Health and Medical

Research (INSERM). G.C. was financially supported by a fellowship from the French Ministry of Scientific Research. P.M. is a senior member of and supported by the Institut Universitaire de France.

## Author contributions

S.-S.H., H.M., G.C., G.S.F., P.B. and P.M. all participated in the design of the experiments. S.-S.H., H.M. and G.C. performed the experiments. S-S.H., H.M., P.B. and P.M. wrote the manuscript.

## Additional information

**Competing interests:** The authors declare no competing financial interests.

