## [Peer Review file · Nature Communications]

Reviewers' comments:

Reviewer #1 (Remarks to the Author):

This project appears to have been conducted in a workman-like manner, but it lacks originality.

Experimenters have been conducting this type of research for decades, using a variety of different transgenes whose products promote apoptosis of synovial cells, including p53, FasL and tk/ganciclovir. Ad5 has been the vector of choice for such experiments. Tom Huizinga showed that the tk/ganciclovir combination was effective in monkeys with collagen-induced arthritis and took this into a human clinical trial for aseptic loosening.

The only novelty of this paper is to use PUMA as the apoptotic gene and a adenovirus-baculovirus combination vector.

Reviewer #2 (Remarks to the Author):

An interesting approach for PUMA gene transfer of the PUMA to synoviocytes combining adenovirus type 5 with a baculovirus carrier was used and its efficiency confirmed in an arthritis animal model. This is a highly relevant topic, however, there are some issues regarding the study:

-An expression system for the PUMA sequence has been designed. Please state whether the human sequence was inserted in methods. This construct efficiently induced apoptosis in human synoviocytes as clearly shown in figure 3. Due to the differences in the PUMA sequence between species, please clarify whether the human construct was used in the animal model. If the human PUMA sequence was used, please show the transduction efficiency and apoptosis induction in rat cells. This would help to estimate the transfer to application in humans.

-The method using the combination of both viruses and the piggybacking approach was excellently described in 2009 (Granio et al., J. Virol, citation #14) clearly showing that these well evaluated constructs are suitable for gene transfer into CAR-negative cells. In fact, Figure part 1A (b) is identical with one of the figure parts in this publication. However, as the transduction efficiency of human synoviocytes with GFP was shown in detail, Figure 1 could be removed with the exception of the schematic model. Figure part 1A (a) could be implemented in Figure 2.

-For human application, depletion of local cartilage-destructive and pro-inflammatory FLS is the major goal of the PUMA-induction into joints. However, it would be of interest which other cells are targeted (surface chondrocytes? Bone cells in the eroded area? Other synovial cells?) e.g. via stainings in the joint tissues or using isolated human/murine cells.

-Duration of the effect: Although additionally to the 4-day approach a 21-day experiment was performed confirming the treatment efficacy after 3 weeks, it would be of interest to show (1) that the effect is local and no viruses leave the injected joints (2) that there is a sufficient depletion of FLS at the site of joint erosion (e.g. by immunohistochemistry at day 21) and (3) how long the viruses remain active/stable and are able to affect potentially still present FLS as rheumatoid arthritis is a lifelong disease. For clinical application, repetitive treatments may prove difficult and repetitive treatments may be necessary. And finally, it would be of interest whether there is a response of the immune system to both viruses.

Specific comments:

- Page 6, line 144ff: It is mentioned that three synovium explants were evaluated with intact cell-cell interactions between FLS and immune cells. However, in Figure 3 isolated FLS are mentioned. Please clarify.

-Please highlight the number of experimental and or biological replicates (samples from different

patients) for each result. Please also mention the p-values and which results/values were significant.

- Figure 3B: Please provide an evaluation of the tree patients evaluated and statistics for the 3 biological replicates.

-In figure 5, control ankles from animals without arthritis are shown in comparison to the treatment arms and non-treated arthritis controls. It would be of interest to include this group in figure 4 to show whether treatment leads to a near-healthy situation after PUMA-application.

-Cell transduction assay: Please clarify 'appropriate volumes' of each virus.

-Please mention how the fresh-frozen sections were prepared (non-decalcified bone tissues?). This information could not be found in citation 25.

-For statistical analysis, each ankle joint was treated independently for statistical purposes. Potentially there may be animal-specific similarities even though the joints were injected individually. Usually, one value per animal is used.

Reviewer #3 (Remarks to the Author):

Hong and colleagues show that PUMA transfection by baculovirus enhances the apoptosis of synovial fibroblasts. In further experiment they show that overexpression of PUMA improves the resolution of antigen-induced arthritis in rats. This is an interesting manuscript showing that induction of apoptosis in the joints fosters the resolution of established arthritis.

Some points need to be considered:

1. The effects of TNF- α and IL-17 stimulation are unclear. There is not difference regarding apoptosis upon PUMA transfection whether cytokines are absent or present. While it is nice to show that PUMA over-expression may induce apoptosis also in synovial fibroblasts under inflammatory conditions, one would like to know whether treatment with these cytokines increases survival /resistance to apoptosis in mock-transfected cells. This would provide at least a rationale to investigate cytokine exposed synovial fibroblasts.
2. Figure 3 is not entirely clear. What is the difference between the cells used in part A and part B. Why did the authors use an apoptosis assay in A and then switch to another assay (MTT) in B?
3. The analysis of the cell number shown in figure 5F is very crude and does not sufficiently characterize what is going on in the joints.
4. Is there apoptosis of synovial cells in the PUMA transfected mice? Is there an intact synovial lining in these animals?
5. It would be interesting to see what happens in the joint? PUMA obviously facilitates resolution of arthritis. Hence neutrophils and T cells as well as monocytes in the inflamed joints may die as well or they may escape from the joint. One guesses that PUMA transfection may also affect cells other than synovial fibroblasts leading to enhanced apoptosis of e.g. immune cells in the joint. The authors need to provide at least some descriptive data on what happens in the joints of mice injected with the PUMA vector.

Reviewer #1 :

This project appears to have been conducted in a workman-like manner, but it lacks originality.

Experimenters have been conducting this type of research for decades, using a variety of different transgenes whose products promote apoptosis of synovial cells, including p53, FasL and tk/ganciclovir. Ad5 has been the vector of choice for such experiments. Tom Huizinga showed that the tk/ganciclovir combination was effective in monkeys with collagen-induced arthritis and took this into a human clinical trial for aseptic loosening.

The only novelty of this paper is to use PUMA as the apoptotic gene and a adenovirus-baculovirus combination vector.

We thank Reviewer 1 for his/her remarks in particular, that he/she finds that using PUMA as the apoptotic gene and an adenovirus-baculovirus combination vector a novelty. We are aware of the previous gene therapy work in arthritis and have contributed to this literature.

There are two key innovative aspects:

1. Use of the PUMA gene as a far more efficient apoptosis-inducing gene, and one that bypasses some of the efficacy issues that have arisen with previous attempts using p53 as a transgene. In fact, the gene is so effective that the vector production is challenging due to death of the producer cell line. Overcoming this issue is another area that would be interesting to the field.
2. The development of an optimized Ad5 vector is probably the key innovation. It has been known for years that transduction efficiency is very limited using standard vectors. We have modified the vector delivery so that it is far more effective and overcomes the need for large volumes of very high titer virus. This also decreases the pro-inflammatory issues that have arisen with using the standard vectors.

Reviewer #2 :

An interesting approach for PUMA gene transfer of the PUMA to synoviocytes combining adenovirus type 5 with a baculovirus carrier was used and its efficiency confirmed in an arthritis animal model. This is a highly relevant topic, however, there are some issues regarding the study:

1. An expression system for the PUMA sequence has been designed. Please state whether the

human sequence was inserted in methods. This construct efficiently induced apoptosis in human synoviocytes as clearly shown in figure 3. Due to the differences in the PUMA sequence between species, please clarify whether the human construct was used in the animal model. If the human PUMA sequence was used, please show the transduction efficiency and apoptosis induction in rat cells. This would help to estimate the transfer to application in humans.

The human PUMA sequence was used and inserted in our Adenovirus vector. This information is now added to the Materials & Methods section, page 16, line 348.

The transduction efficiency in rat FLS with our double vector system is now shown in Fig. 1B and described in page 5, line 110-114. The apoptosis induction in rat cells are shown in (Fig. 2i, j) and described in page 6, line 128-133. The high efficiency in rat cells allowed to use the same human-based vector in the rat model.

2. The method using the combination of both viruses and the piggybacking approach was excellently described in 2009 (Granio et al., J. Virol, citation #14) clearly showing that these well evaluated constructs are suitable for gene transfer into CAR-negative cells. In fact, Figure part 1A (b) is identical with one of the figure parts in this publication. However, as the transduction efficiency of human synoviocytes with GFP was shown in detail, Figure 1 could be removed with the exception of the schematic model. Figure part 1A (a) could be implemented in Figure 2.

We thank the reviewer for his/her compliments regarding our previous work, and for alerting us about the duplication of the EM image. We have now substituted the EM image with another which has not been used in any publication.

As suggested, we have removed the photo of the transduction of human synoviocytes, and we have added a new Figure 1B, showing the comparison of transduction efficiency in human and rat synoviocytes, using our vector system.

3. For human application, depletion of local cartilage-destructive and pro-inflammatory FLS is the major goal of the PUMA-induction into joints. However, it would be of interest which other cells are targeted (surface chondrocytes? Bone cells in the eroded area? Other synovial cells?) e.g. via stainings in the joint tissues or using isolated human/murine cells.

As suggested, we have now provided a histological analysis showing protective effects of BV^{CAR}HAAdV5-PUMA on cartilage and bone cells and structure (Fig 6).

4. Duration of the effect: Although additionally to the 4-day approach a 21-day experiment was performed confirming the treatment efficacy after 3 weeks, it would be of interest to show **(1) that the effect is local and no viruses leave the injected joints (2) that there is a sufficient depletion of FLS at the site of joint erosion (e.g. by immunohistochemistry at day 21) and (3) how long the viruses remain active/stable** and are able to affect potentially still present FLS as rheumatoid arthritis is a lifelong disease. For clinical application, repetitive treatments may prove difficult and repetitive treatments may be necessary. And finally, it would be of interest whether there is a response of the immune system to both viruses.

(1) Show that the effect is local and no viruses leave the injected joints.

PCR assays were performed on total DNA extracted from liver and serum samples from the animals treated for 4 days and 21 days. No amplification of adenovirus or baculovirus genome were obtained from the liver or serum samples. These results (data not shown) implied that there was no dissemination of the vectors from the injected joints and is now mentioned in the Results section, page 11, line 242-247.

(2) Histo-morphometry of day 21 treated animals

In the previous version, we had provided a quantification of the cells present in the joints of 4-day treated animals, showing a reduction of both FLS and immune cells (Figure 5F). In addition to this, an extensive histo-morphometry study of the ankle joints of 21-day animals was performed, comparing those treated with BV^{CAR}HAdV5-PUMA and HAdV5-PUMA. We observed a significant reduction of osteoclast staining in the BV^{CAR}HAdV5-PUMA compared to HAdV5-PUMA (Fig 7B, a). We also observed a high amount of mineralized bone BV^{CAR}HAdV5-PUMA compared to HAdV5-PUMA (Fig 7B, b). And the inflammatory infiltrate and cartilage alteration were reduced in BV^{CAR}HAdV5-PUMA compared to HAdV5-PUMA (Fig 7B, c).

This is now described in Page 10, line 229-239, and shown in (Fig. 7B, a, b, c).

(3) Virus persistence - how long the viruses remain active/stable ?

To check for viral persistence post-treatment, DNA was extracted from the synovial tissue of the animals treated with HAdV5-PUMA and BV^{CAR}HAdV5-PUMA for 21 days. PCR assays made on the samples showed the presence of only the adenovirus vector genomes in the synovial tissue. The results are now described in page 11, line 248-256, and shown in (Fig. 8A, a, b).

(4) Were there response of the immune system to both viruses ?

To answer this question, we checked by ELISA assays for the presence of antibodies to adenovirus and baculovirus in the sera of the rats treated with HAdV5-PUMA and BV^{CAR}HAdV5-PUMA for 4 days and 21 days. The results showed that none or insignificant antibodies were detected against adenovirus and baculovirus in the sera of the 4 day treated rats. At 21 days, anti-Ad antibodies were detected while presence of anti-baculovirus antibodies not detectable. We suggest that the immunization occurs mainly inside the joint. The results are now described in page 11, line 257-264, and shown in (Fig. 8B, a, b).

Specific comments:

- Page 6, line 144: It is mentioned that three synovium explants were evaluated with intact cell-cell interactions between FLS and immune cells. However, in Figure 3 isolated FLS are mentioned. Please clarify.

-Please highlight the number of experimental and or biological replicates (samples from different patients) for each result. Please also mention the p-values and which results/values were significant.

- Figure 3B: Please provide an evaluation of the three patients evaluated and statistics for the 3 biological replicates.

We apologise for this confusion. In this experiment, we used FLS derived from RA synovium samples from 3 different patients. The results presented are individual experiments to show the reproducibility between patients. The evaluation of the sensitivity of the 3 different FLS is now mentioned in page 6, line 144. The statistics are now added to Fig. 3A, and mentioned in the legend, page 26, line 609-611.

-In figure 5, control ankles from animals without arthritis are shown in comparison to the treatment arms and non-treated arthritis controls. It would be of interest to include this group in figure 4 to show whether treatment leads to a near-healthy situation after PUMA-application.

As mentioned in the manuscript, the negative control groups were not allowed by our animal ethic committee due to the high level of disease severity at 21 days. For this reason, we

compared two therapeutic approaches to demonstrate the better efficacy of the combined vectors.

-Cell transduction assay: Please clarify 'appropriate volumes' of each virus.

We apologize for the vague description of the cell transduction assay. This is now rectified and the assay is now described in detail in page 17, line 367-371.

-Please mention how the fresh-frozen sections were prepared (non-decalcified bone tissues?). This information could not be found in citation 25.

After dissection of the joints, they were out in OCT and flash frozen in liquid nitrogen. The frozen blocks were then set up in the cryostat to obtain cryosections at 10 μ m. This is now mentioned in Page 22, line 478.

-For statistical analysis, each ankle joint was treated independently for statistical purposes. Potentially there may be animal-specific similarities even though the joints were injected individually. Usually, one value per animal is used.

We apologize for this error and we have performed again the analysis with one value (average of two ankles) per animal. The results have been updated without big changes and reached a similar conclusion.

Reviewer #3 (Remarks to the Author):

Hong and colleagues show that PUMA transfection by baculovirus enhances the apoptosis of synovial fibroblasts. In further experiment, they show that overexpression of PUMA improves the resolution of antigen-induced arthritis in rats. This is an interesting manuscript showing that induction of apoptosis in the joints fosters the resolution of established arthritis.

Some points need to be considered:

1. The effects of TNF- α and IL-17 stimulation are unclear. There is not difference regarding apoptosis upon PUMA transfection whether cytokines are absent or present. While it is nice to show that PUMA over-expression may induce apoptosis also in synovial fibroblasts under inflammatory conditions, one would like to know whether treatment with these cytokines increases survival /resistance to apoptosis in mock-transfected cells. This would provide at least a rationale to investigate cytokine exposed synovial fibroblasts.

Work by us and others have shown that these cytokines induced the over-expression of anti-apoptotic genes in synoviocytes. For instance, IL-17 induces synoviolin expression and protects cells from apoptosis (reference 19, Toh ML et al.). Thus, it could be expected that the vector efficacy would be reduced in cells made more resistant by exposure to inflammation. In contrast, the opposite was observed with no reduction of apoptosis with inflammation. This is very important for the clinical application. This point has been added to the discussion.

2. Figure 3 is not entirely clear. What is the difference between the cells used in part A and part B. Why did the authors use an apoptosis assay in A and then switch to another assay (MTT) in B?

This joins the remark of Reviewer #2 (specific comments). We apologize for the confusion. To make it clearer to the reader, we have inverted the presentation of the previous Fig 3A and Fig 3B.

In the new Fig. 3A, the MTT test was first used to compare the sensitivity of FLS from 3

different RA patients to PUMA-induced cell death.

In the new Fig. 3B, an ELISA assay to measure the degree of nuclear fragmentation was performed on one of the three FLS in Fig 3A to confirm that the observed cell death is due to apoptosis. This is now explained in page 7, line 154-156.

3. The analysis of the cell number shown in figure 5F is very crude and does not sufficiently characterize what is going on in the joints.

We have added a representative photo of each group, in the new Fig. 6.

4. Is there apoptosis of synovial cells in the PUMA transfected mice?

The reviewer must be referring to “rat” and not “mice” synovial cells, which joins the Point 1 of Reviewer #2. The efficiency of PUMA-induced apoptosis in rat cells are shown in (Fig. 2 i, j) and described in page 6, line 128-133. This allows the use of a rat model with this human-based vector.

Is there an intact synovial lining in these animals?

Data from the histo-morphometry (Fig. 7B) suggested integrity of the synovial lining in BV^{CAR}HAdV5-PUMA compared to HAdV5-PUMA.

5. It would be interesting to see what happens in the joint? PUMA obviously facilitates resolution of arthritis. Hence neutrophils and T cells as well as monocytes in the inflamed joints may die as well or they may escape from the joint. One guesses that PUMA transfection may also affect cells other than synovial fibroblasts leading to enhanced apoptosis of e.g. immune cells in the joint. The authors need to provide at least some descriptive data on what happens in the joints of mice injected with the PUMA vector.

We have now provided additional data on histo-morphometry showing a strong effect and reduction in numbers of osteoclasts and immune cell infiltrate in BV^{CAR}HAdV5-PUMA compared to HAdV5-PUMA (Fig. 7B, a, b,).

REVIEWERS' COMMENTS:

Reviewer #2 (Remarks to the Author):

Comments 1, 2, 4(1-3) as well as the specific comments were sufficiently addressed and additional information provided.

Comment 3:

>>>For human application, depletion of local cartilage-destructive and pro-inflammatory FLS is the major goal of the PUMA-induction into joints. However, it would be of interest which other cells are targeted (surface chondrocytes? Bone cells in the eroded area? Other synovial cells?) e.g. via stainings in the joint tissues or using isolated human/murine cells.

Answer: As suggested, we have now provided a histological analysis showing protective effects of BVCARHAdV5-PUMA on cartilage and bone cells and structure (Fig 6).<<<

In Fig. 6, the left panel giving an overview could be removed and instead insets showing a higher magnification included.

In Fig. 7, the joint/bone structures are nicely shown. Smaller arrows could be used as well as some insets showing the key findings. In part, the areas the arrows are pointing at are difficult to see.

Comment 4(4):

>>>(4) Were there response of the immune system to both viruses ?

Answer: To answer this question, we checked by ELISA assays for the presence of antibodies to adenovirus and baculovirus (...) At 21 days, anti-Ad antibodies were detected while presence of anti-baculovirus antibodies not detectable. We suggest that the immunization occurs mainly inside the joint. The results are now described in page 11, line 257-264, and shown in (Fig. 8B, a, b).<<<

A section should be added discussing these findings with regard to application for human therapy.

Reviewer #3 (Remarks to the Author):

The authors have addressed all my questions.

We have addressed the three additional points raised by Reviewer #2.

Reviewer#2, Point 1 : In Fig. 6, the left panel giving an overview could be removed and instead insets showing a higher magnification included.

This modification is now done for Fig.6.

Reviewer#2, Point 2 :

In Fig. 7, the joint/bone structures are nicely shown. Smaller arrows could be used as well as some insets showing the key findings. In part, the areas the arrows are pointing at are difficult to see.

We have now reduced the size of the arrows in Fig.7.

Reviewer#2, Point 3 :

Were there response of the immune system to both viruses ?

A section should be added discussing these findings with regard to application for human therapy.

We have now discussed this point in the Discussion.

We are very grateful to Reviewer#2 for his/her positive comments and suggestions, which have helped further improved our manuscript.